# Temporal Efficient Training of Spiking Neural Network via Gradient Re-weighting

**Shikuang Deng**[1,2]**, Yuhang Li**[3]**, Shanghang Zhang**[4] **& Shi Gu**[1,2,5✉]
[1]University of Electronic Science and Technology of China,
[2]Shenzhen Institute for Advanced Study, UESTC
[3]Yale University, [4]Peking University ,[5]Peng Cheng Laboratory
`dengsk119@std.uestc.edu.cn, yuhang.li@yale.edu, gus@uestc.edu.cn`

## Abstract

Recently, brain-inspired spiking neuron networks (SNNs) have attracted widespread research interest because of their event-driven and energy-efficient characteristics. Still, it is difficult to efficiently train deep SNNs due to the non-differentiability of its activation function, which disables the typically used gradient descent approaches for traditional artificial neural networks (ANNs). Although the adoption of surrogate gradient (SG) formally allows for the back-propagation of losses, the discrete spiking mechanism actually differentiates the loss landscape of SNNs from that of ANNs, failing the surrogate gradient methods to achieve comparable accuracy as for ANNs. In this paper, we first analyze why the current direct training approach with surrogate gradient results in SNNs with poor generalizability. Then we introduce the temporal efficient training (TET) approach to compensate for the loss of momentum in the gradient descent with SG so that the training process can converge into flatter minima with better generalizability. Meanwhile, we demonstrate that TET improves the temporal scalability of SNN and induces a temporal inheritable training for acceleration. Our method consistently outperforms the SOTA on all reported mainstream datasets, including CIFAR-10/100 and ImageNet. Remarkably on DVS-CIFAR10, we obtained 83% top-1 accuracy, over 10% improvement compared to existing state of the art. Codes are available at https://github.com/Gus-Lab/temporal_efficient_training.

## 1 Introduction

The advantages of Spiking neuron networks (SNNs) lie in their energy-saving and fast-inference computation when embedded on neuromorphic hardware such as TrueNorth (DeBole et al., 2019) and Loihi (Davies et al., 2018). Such advantages originate from the biology-inspired binary spike transmitted mechanism, by which the networks avoid multiplication during inference. On the other hand, this mechanism also leads to difficulty in training very deep SNNs from scratch because the non-differentiable spike transmission hinders the powerful back-propagation approaches like gradient descents. Recently, many studies on converting artificial neuron networks (ANNs) to SNNs have demonstrated SNNs' comparable power in feature representation as ANNs (Han & Roy, 2020; Deng & Gu, 2020; Li et al., 2021a). Nevertheless, it is commonly agreed that the direct training method for high-performance SNN is still crucial since it distinguishes SNNs from converted ANNs, especially on neuromorphic datasets.

The output layer's spike frequency or the average membrane potential increment is commonly used as inference indicators in SNNs (Shrestha & Orchard, 2018; Kim et al., 2019). The current standard direct training (SDT) methods regard the SNN as RNN and optimize inference indicators' distribution (Wu et al., 2018). They adopt surrogate gradients (SG) to relieve the non-differentiability (Lee et al., 2016; Wu et al., 2018; Zheng et al., 2021). However, the gradient descent with SG does not match with the loss landscape in SNN and is easy to get trapped in a local minimum with low generalizability. Although using suitable optimizers and weight decay help ease this problem, the

---

✉ Corresponding author

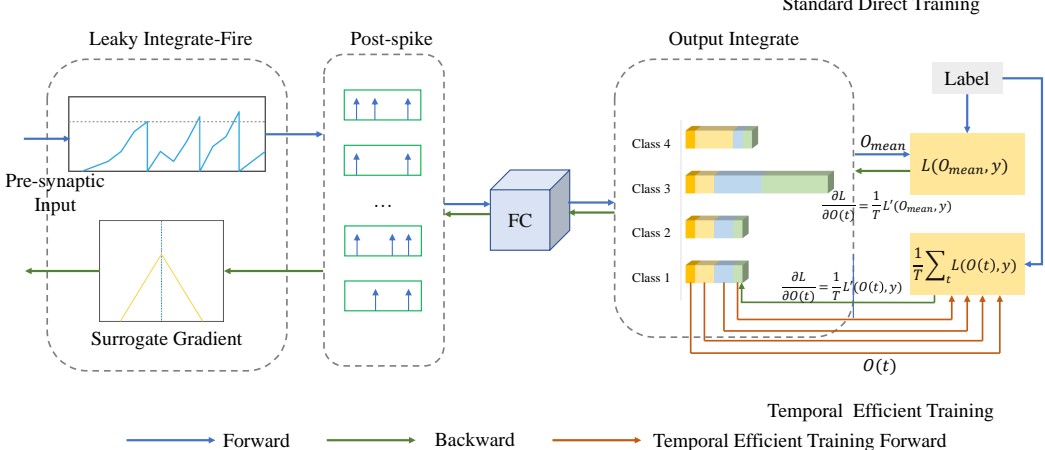

Figure 1: Workflow of temporal efficient training (TET). To obtain a more generalized SNN, we modify the optimization target to adjust each moment's output distribution.

performance of deep SNNs trained from scratch still suffers a big deficit compared to that of ANNs Deng et al. (2020). Another training issue is the memory and time consumption, which increases linearly with the simulation time. Rathi & Roy (2020) initializes the target network by a converted SNN to shorten the training epochs, indicating the possibility of high-performance SNN with limited activation time. The training problem due to the non-differentiable activation function has become the main obstruction of spiking neural network development.

In this work, we examine the limitation of the traditional direct training approach with SG and propose the temporal efficient training (TET) algorithm. Instead of directly optimizing the integrated potential, TET optimizes every moment's pre-synaptic inputs. As a result, it avoids the trap into local minima with low prediction error but a high second-order moment. Furthermore, since the TET applies optimization on each time point, the network naturally has more robust time scalability. Based on this characteristic, we propose the time inheritance training (TIT), which reduces the training time by initializing the SNN with a smaller simulation length. With the help of TET, the performance of SNNs has improved on both static datasets and neuromorphic datasets. Figure 1 depicts the workflow of our approach.

The following summarizes our main contributions:

- We analyze the problem of training SNN with SG and propose the TET method, a new loss and gradient descent regime that succeeds in obtaining more generalizable SNNs.
- We analyze the feasibility of TET and picture the loss landscape under both the SDT and TET setups to demonstrate TET's advantage in better generalization.
- Our sufficient experiments on both static datasets and neuromorphic datasets prove the effectiveness of the TET method. Especially on DVS-CIFAR10, we report $83.17\%$ top-1 accuracy for the first time, which is over $10\%$ better than the current state-of-the-art result.

## 2 RELATED WORK

In recent years, SNNs have developed rapidly and received more and more attention from the research community. However, lots of challenging problems remain to be unsolved. In general, most works on SNN training have been carried out in two strategies: ANN-to-SNN conversion and direct training from scratch.

**ANN-to-SNN Conversion.** Conversion approaches avoid the training problem by trading high accuracy through high latency. They convert a high-performing ANN to SNN and adjust the SNN parameters w.r.t the ANN activation value layer-by-layer (Diehl et al., 2015; 2016). Some special techniques have been proposed to reduce the inference latency, such as the subtraction mechanism

(Rueckauer et al., 2016; Han et al., 2020), robust normalization Rueckauer et al. (2016), spike-norm (Sengupta et al., 2018), and channel-wise normalization (Kim et al., 2019). Recently, Deng & Gu (2020) decompose the conversion error to each layer and reduce it by bias shift. Li et al. (2021a) suggest using adaptive threshold and layer-wise calibration to obtain high-performance SNNs that require a simulation length of less than 50. However, converted methods significantly extend the inference latency, and they are not suitable for neuromorphic data (Deng et al., 2020).

**Direct training.** In this area, SNNs are regarded as special RNNs and training with BPTT (Neftci et al., 2019). On the backpropagation process, The non-differentiable activation term is replaced with a surrogate gradient (Lee et al., 2016). Compared with ANN-to-SNN conversion, direct training achieves high accuracy with few time steps but suffers more training costs (Deng et al., 2020). Several studies suggest that surrogate gradient (SG) is helpful to obtain high-performance SNNs on both static datasets and neuromorphic datasets (Wu et al., 2019; Shrestha & Orchard, 2018; Li et al., 2021b). On the backpropagation process, SG replaces the Dirac function with various shapes of curves. Exceptionally, Wu et al. (2018) first propose the STBP method and train SNNs on the ANN programming platform, which significantly promotes direct training development. Zheng et al. (2021) further proposes the tdBN algorithm to smooth the loss function and first realize training a large-scale SNN on ImageNet. Zhang & Li (2020) proposes TSSL-BP to break down error backpropagation across two types of inter-neuron and intra-neuron dependencies and achieve low-latency and high accuracy SNNs. Recently, Yang et al. (2021) designed a neighborhood aggregation (NA) method to use the multiple perturbed membrane potential waveforms in the neighborhood to compute the finite difference gradients and guide the weight updates. They significantly decrease the required training iterations and improve the SNN performance.

## 3 PRELIMINARY

### 3.1 ITERATIVE LIF MODEL

We adopt the Leaky Integrate-and-Fire (LIF) model and translate it to an iterative expression with the Euler method (Wu et al., 2019). Mathematically, the membrane potential is updated as

$$\boldsymbol{u}(t+1) = \tau \boldsymbol{u}(t) + \boldsymbol{I}(t), \tag{1}$$

where $\tau$ is the constant leaky factor, $\boldsymbol{u}(t)$ is the membrane potential at time $t$, and $\boldsymbol{I}(t)$ denotes the pre-synaptic inputs, which is the product of synaptic weight $\mathbf{W}$ and spiking input $\boldsymbol{x}(t)$. Given a specific threshold $V_{th}$, the neuron fires a spike and $\boldsymbol{u}(t)$ reset to 0 when the $\boldsymbol{u}(t)$ exceeds the threshold. So the firing function and hard reset mechanism can be described as

$$\boldsymbol{a}(t+1) = \boldsymbol{\Theta}(\boldsymbol{u}(t+1) - V_{th}) \tag{2}$$

$$\boldsymbol{u}(t+1) = \boldsymbol{u}(t+1) \cdot (1 - \boldsymbol{a}(t+1)), \tag{3}$$

where $\boldsymbol{\Theta}$ denotes the Heaviside step function. The output spike $\boldsymbol{a}(t+1)$ will become the post synaptic spike and propagate to the next layer. In this study, we set the starting membrane $\boldsymbol{u}(0)$ to 0, the threshold $V_{th}$ to 1, and the leaky factor $\tau$ to 0.5 for all experiments.

The last layer's spike frequency is typically used as the final classification index. However, adopting the LIF model on the last layer will lose information on the membrane potential and damage the performance, especially on complex tasks (Kim et al., 2019). Instead, we integrate the pre-synaptic inputs $\boldsymbol{I}(t)$ with no decay or firing (Rathi & Roy, 2020; Fang et al., 2021). Finally, we set the average membrane potential as the classification index and calculate the cross-entropy loss for training.

### 3.2 SURROGATE GRADIENT

Following the concept of direct training, we regard the SNN as RNN and calculate the gradients through spatial-temporal backpropagation (STBP) (Wu et al., 2018):

$$\frac{\partial L}{\partial \mathbf{W}} = \sum_t \frac{\partial L}{\partial \boldsymbol{a}(t)} \frac{\partial \boldsymbol{a}(t)}{\partial \boldsymbol{u}(t)} \frac{\partial \boldsymbol{u}(t)}{\partial \boldsymbol{I}(t)} \frac{\partial \boldsymbol{I}(t)}{\partial \mathbf{W}}, \tag{4}$$

where the term $\frac{\partial \boldsymbol{a}(t)}{\partial \boldsymbol{u}(t)}$ is the gradient of the non-differentiability step function involving the derivative of Dirac's $\delta$-function that is typically replaced by surrogate gradients with a derivable curve. So far,

there are various shapes of surrogate gradients, such as rectangular (Wu et al., 2018; 2019), triangle (Esser et al., 2016; Rathi & Roy, 2020), and exponential (Shrestha & Orchard, 2018) curve. In this work, we choose the surrogate gradients shaped like triangles. Mathematically, it can describe as

$$\frac{\partial \boldsymbol{a}(t)}{\partial \boldsymbol{u}(t)} = \frac{1}{\gamma^2} \max(0, \gamma - |\boldsymbol{u}(t) - V_{th}|), \tag{5}$$

where the $\gamma$ denotes the constraint factor that determines the sample range to activate the gradient.

### 3.3 BATCH NORMALIZATION FOR SNN

Batch Normalization (BN) (Ioffe & Szegedy, 2015) is beneficial to accelerate training and increase performance since it can smooth the loss landscape during training (Santurkar et al., 2018). Zheng et al. (2021) modified the forward time loop form and proposed threshold-dependent Batch Normalization (tdBN) to normalize the pre-synaptic inputs $\boldsymbol{I}$ in both spatial and temporal paradigms so that the BN can support spatial-temporal input. We adopt this setup with the extension of the time dimension to batch dimension [1]. In the inference process, the BN layer will be merged into the pre-convolutional layer, thus the inference rule of SNN remain the same but with modified weight:

$$\hat{\mathbf{W}} \leftarrow \mathbf{W}\frac{\gamma}{\alpha}, \hat{\boldsymbol{b}} \leftarrow \beta + (\boldsymbol{b} - \mu)\frac{\gamma}{\alpha}, \tag{6}$$

where $\mu, \alpha$ are the running mean and standard deviation on both spatial and temporal paradigm, $\gamma, \beta$ are the affine transformation parameters, and $\mathbf{W}, \boldsymbol{b}$ are the parameters of the pre-convolutional layer.

## 4 METHODOLOGY

### 4.1 FORMULA OF TRAINING SNN WITH SURROGATE GRADIENTS

**Standard Direct Training.** We use $\boldsymbol{O}(t)$ to represent pre-synaptic input $\boldsymbol{I}(t)$ of the output layer and calculate the cross-entropy loss. The loss function of standard direct training $\mathcal{L}_{\text{SDT}}$ is:

$$\mathcal{L}_{\text{SDT}} = \mathcal{L}_{\text{CE}}(\frac{1}{T}\sum_{t=1}^{T} \boldsymbol{O}(t), \boldsymbol{y}), \tag{7}$$

where $T$ is the total simulation time, $\mathcal{L}_{\text{CE}}$ denotes the cross-entropy loss, and $\boldsymbol{y}$ represents the target label. Following the chain rule, we obtain the gradient of $\mathbf{W}$ with softmax $S(\cdot)$ inference function :

$$\frac{\partial \mathcal{L}_{\text{SDT}}}{\partial \mathbf{W}} = \frac{1}{T}\sum_{t=1}^{T}[S(\boldsymbol{O}_{\text{mean}}) - \hat{\boldsymbol{y}}]\frac{\partial \boldsymbol{O}(t)}{\partial \mathbf{W}}, \tag{8}$$

where $\boldsymbol{O}_{\text{mean}}$ denotes the average of the output $\boldsymbol{O}(t)$ over time, and $\hat{\boldsymbol{y}}$ is the one-hot coding of $\boldsymbol{y}$.

**Temporal Efficient Training.** In this section, we come up with a new kind of loss function $\mathcal{L}_{\text{TET}}$ to realize temporal efficient training (TET). It constrains the output (pre-synaptic inputs) at each moment to be close to the target distribution. It is described as:

$$\mathcal{L}_{\text{TET}} = \frac{1}{T} \cdot \sum_{t=1}^{T} \mathcal{L}_{\text{CE}}[\boldsymbol{O}(t), \boldsymbol{y}]. \tag{9}$$

Recalculate the gradient of weights under the loss function $\mathcal{L}_{\text{TET}}$, and we have:

$$\frac{\partial \mathcal{L}_{\text{TET}}}{\partial \mathbf{W}} = \frac{1}{T}\sum_{t=1}^{T}[S(\boldsymbol{O}(t)) - \hat{\boldsymbol{y}}] \cdot \frac{\partial \boldsymbol{O}(t)}{\partial \mathbf{W}}. \tag{10}$$

---

[1] https://github.com/fangwei123456/spikingjelly

## 4.2 CONVERGENCE OF GRADIENT DESCENT FOR SDT V.S. TET

In the case of SDT, the gradient consists of two parts, the error term $(S(\boldsymbol{O}_{\text{mean}}) - \hat{\boldsymbol{y}})$ and the partial derivative of output $\partial \boldsymbol{O}(t)/\partial \mathbf{W}$. When the training process reaches near a local minimum, the term $(S(\boldsymbol{O}_{\text{mean}}) - \hat{\boldsymbol{y}})$ approximates $\mathbf{0}$ for all $t = 1, ..., T$, ignorant of the term $\partial \boldsymbol{O}(t)/\partial \mathbf{W}$. For traditional ANNs, the accumulated momentum may help get out of the local minima (e.g. saddle point) that typically implies bad generalizability (Kingma & Ba, 2014; Kidambi et al., 2018). However, when the SNN is trained with surrogate gradients, the accumulated momentum could be extremely small, considering the mismatch of gradients and losses. The fact that the activation function is a step one while the SG is bounded with integral constraints. This mismatch dissipates the momentum around a local minimum and stops the SDT from searching for a flatter minimum that may suggest better generalizability.

In the case of TET, this issue of mismatch is relieved by reweighting the contribution of $\partial \boldsymbol{O}(t)/\partial \mathbf{W}$. Indeed, considering the fact that the first term $(S(\boldsymbol{O}(t)) - \hat{\boldsymbol{y}})$ is impossible to be $\mathbf{0}$ at every moment of SNN since the early output accuracy on the training set is not $100\%$. So TET needs the second term $\partial \boldsymbol{O}(t)/\partial \mathbf{W}$ close to 0 to make the $\mathcal{L}_{\text{TET}}$ convergence. This mechanism increases the norm of gradients around sharp local minima and drives the TET to search for a flat local minimum where the disturbance of weight does not cause a huge change in $\boldsymbol{O}(t)$.

Further, to ensure that the convergence with TET implies the convergence of SDT, we prove the following lemma:

**Lemma 4.1.** $\mathcal{L}_{SDT}$ *is upper bounded by* $\mathcal{L}_{TET}$.

*Proof.* Suppose $\boldsymbol{O}_i(t)$ and $\hat{\boldsymbol{y}}_i$ denote the i-th component of $\boldsymbol{O}(t)$ and $\hat{\boldsymbol{y}}$, respectively. Expand Eqn.9, we have:

$$
\begin{aligned}
\mathcal{L}_{\text{TET}} &= -\frac{1}{T}\sum_{t=1}^{T}\sum_{i=1}^{n}\hat{\boldsymbol{y}}_i \log S(\boldsymbol{O}_i(t)) = -\frac{1}{T}\sum_{i=1}^{n}\hat{\boldsymbol{y}}_i \log(\prod_{t=1}^{T} S(\boldsymbol{O}_i(t))) \\
&= -\sum_{i=1}^{n}\hat{\boldsymbol{y}}_i \log(\prod_{t=1}^{T} S(\boldsymbol{O}_i(t)))^{\frac{1}{T}} \geq -\sum_{i=1}^{n}\hat{\boldsymbol{y}}_i \log(\frac{1}{T}\sum_{t=1}^{T} S(\boldsymbol{O}_i(t))) \\
&\geq -\sum_{i=1}^{n}\hat{\boldsymbol{y}}_i \log(S(\frac{1}{T}\sum_{t=1}^{T}\boldsymbol{O}_i(t))) = \mathcal{L}_{\text{SDT}},
\end{aligned}
\tag{11}
$$

where the first inequality is given by the Arithmetic Mean-Geometric Mean Inequality, and the second one is given by Jensen Inequality since the softmax function is convex. As a corollary, once the $\mathcal{L}_{\text{TET}}$ gets closed to zero, the original loss function $\mathcal{L}_{\text{SDT}}$ also approaches zero. □

Furthermore, the network output $\boldsymbol{O}(t)$ at a particular time point may be a particular outlier that dramatically affects the total output since the output of the SNN has the same weight at every moment under the rule of integration. Thus it is necessary to add a regularization term like $\mathcal{L}_{\text{MSE}}$ loss to confine each moment's output to reduce the risk of outliers:

$$
\mathcal{L}_{\text{MSE}} = \frac{1}{T}\sum_{t=1}^{T}\text{MSE}(\mathbf{O}(t), \phi),
\tag{12}
$$

where $\phi$ is a constant used to regularize the membrane potential distribution. And we set $\phi = V_{th}$ in our experiments. In practice, we use a hyperparameter $\lambda$ to adjust the proportion of the regular term, we have:

$$
\mathcal{L}_{\text{TOTAL}} = (1 - \lambda)\mathcal{L}_{\text{TET}} + \lambda\mathcal{L}_{\text{MSE}}.
\tag{13}
$$

It is worth noting that we only changed the loss function in the training process and did not change SNN's inference rules in the testing phase for a fair comparison. This algorithm is detailed in Algo.1.

---

**Algorithm 1:** Temporal efficient training for one epoch

---

**Input:** SNN model; Simulation length: $T$; Threshold: $V_{th}$; Training dataset; Validation dataset; total training iteration in one epoch: $I_{train}$; total validation iteration in one epoch: $I_{val}$

**for** *all* $i = 1, 2, ... I_{train}$ *iteration* **do**

    Get mini-batch training data, and class label: $\boldsymbol{Y}^i$;

    Compute the SNN output $\boldsymbol{O}^i(t)$ of eatch time step;

    Calculate loss function: $\mathcal{L}_{\text{TOTAL}} = (1 - \lambda)\mathcal{L}_{\text{TET}} + \lambda\mathcal{L}_{\text{MSE}} =$

    $(1 - \lambda) \cdot \frac{1}{T}\sum_{t=1}^{T}\mathcal{L}_{CE}(\boldsymbol{O}^i(t), \boldsymbol{Y}^i) + \lambda \cdot \frac{1}{T}\sum_{t=1}^{T}\text{MSE}(\boldsymbol{O}^i(t), \phi)$;

    Backpropagation and update model parameters;

**end**

**for** *all* $i = 1, 2, ... I_{val}$ *iteration* **do**

    Get mini-batch validation data, and class label: $\boldsymbol{Y}^i$;

    Compute the SNN average output $\boldsymbol{O}^i_{\text{mean}} = \frac{1}{T}\sum_{t=1}^{T}\boldsymbol{O}^i(t)$ over all time step;

    Compare the classification factor $\boldsymbol{O}^i_{\text{mean}}$ and $\boldsymbol{Y}^i$ for classification;

**end**

---

### 4.3 TIME INHERITANCE TRAINING

SNN demands simulation length long enough to obtain a satisfying performance, but the training time consumption will increase linearly as the simulation length grows. So how to shorten the training time is also an essential problem in the direct training field. Traditional loss function $\mathcal{L}_{\text{SDT}}$ only optimizes the whole network output under a specific $T$, so its temporal scalability is poor. Unlike the standard training, TET algorithm optimizes each moment's output, enabling us to extend the simulation time naturally. We introduce Time Inheritance Training (TIT) to alleviate the training time problem. We first use long epochs to train an SNN with a short simulation time T, e.g., 2. Then, we increase the simulation time to the target value and retrain with short epochs. We discover that TIT performs better than training from scratch on accuracy and significantly saves the training time. Assuming that training an SNN with simulation length $T = 1$ cost $ts$ time per epoch, the SNN needs 300 epochs to train from scratch, and the TIT needs 50 epochs for finetuning. So we need $1800ts$ time to train an SNN with $T = 6$ from scratch, but following the TIT pipeline with the initial $T = 2$ only requires $900ts$. As a result, the TIT can reduce the training time cost by half.

## 5 EXPERIMENTS

We validate our proposed TET algorithm and compare it with existing works on both static and neuromorphic datasets. The network architectures in this paper include ResNet-19 (Zheng et al., 2021), Spiking-ResNet34 (Zheng et al., 2021), SEW-ResNet34 (Fang et al., 2021), SNN-5, and VGGSNN. SNN-5 (16C3-64C5-AP2-128C5-AP2-256C5-AP2-512C3-AP2-FC) is a simple convolutional SNN suitable for multiple runs to discover statistical rules (Figure A. 7). The architecture of VGGSNN (64C3-128C3-AP2-256C3-256C3-AP2-512C3-512C3-AP2-512C3-512C3-AP2-FC) is based on VGG11 with two fully connected layers removed as we found that additional fully connected layers were unnecessary for neuromorphic datasets.

### 5.1 MODEL VALIDATION AND ABLATION STUDY

**Effectiveness of TET over SDT with SG.** We first examine whether the mismatch between SG and loss causes the convergence problem. For this purpose, we set the simulation length to 4 and change the spike function $\Theta$ in Eqn.2 to Sigmoid $\sigma(k \cdot \text{input})$. We find that the TET and SDT achieved similar accuracy (Table 2) when $k = 1, 10, 20$. This indicates that both TET and SDT work when the gradient and loss function match each other. Next, we compare the results training with $\mathcal{L}_{\text{SDT}}$ and $\mathcal{L}_{\text{TET}}$ on SNNs (ResNet-19 on CIFAR100) training with surrogate gradient for three runs. As shown in Table 1, our proposed new TET training strategy dramatically increases the accuracy by 3.25% when the simulation time is 4 and 3.53% when the simulation time is 6. These results quantitatively support the effectiveness of TET in solving the mismatch between gradient and loss in training SNNs with SG.

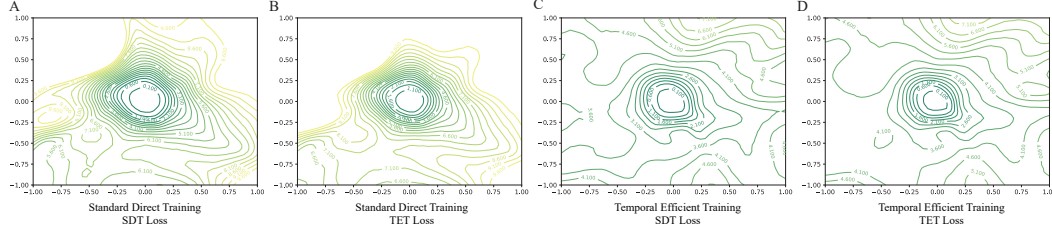

Figure 2: Loss landscape of VGGSNN. The 2D landscape of $\mathcal{L}_{\text{SDT}}$ and $\mathcal{L}_{\text{TET}}$ from two different training methods.

Table 1: Comparison between SDT and TET. We adopt the SNN architecture ResNet-19 with SG on CIFAR100 and record the results with three different simulation lengths 2, 4, and 6.

| Method | T=2 | T=4 | T = 6 |
|---|---|---|---|
| Direct training | 69.41±0.08 | 70.86±0.22 | 71.12±0.57 |
| TET | 72.37±0.21 | 74.11±0.18 | 74.65±0.12 |

Table 2: Comparison of SDT and TET with sigmoid function $\sigma(k\cdot\text{input})$. We fix the simulation length to 4 and record the results of CNN-5 under three different $k$ on CIFAR10.

| Method | k=1 | k=10 | k=20 |
|---|---|---|---|
| Direct training | 88.00±0.15 | 88.83±0.32 | 88.50±0.32 |
| TET | 87.63±0.38 | 89.31±0.15 | 88.64±0.28 |

**Loss Landscape around Local Minima.** We further inspect the 2D landscapes (Li et al., 2018) of $\mathcal{L}_{\text{SDT}}$ and $\mathcal{L}_{\text{TET}}$ around their local minima (see Figure. 2) to demonstrate why TET generalizes better than SDT and how TET helps the training process jump out of the sharp local minima typically found by SDT. First, comparing Figure. 2 A and C, we can see that although the values of local minima achieved by SDT and TET are similar in $\mathcal{L}_{\text{SDT}}$, the local minima of TET (Figure. 2 C) is flatter than that of SDT (Figure. 2 A). This indicates that the TET is effective in finding flatter minima that are typically more generalizable even w.r.t the original loss in TET. Next, we examine the two local minima under $\mathcal{L}_{\text{TET}}$ to see how it helps jump out the local minima found by SDT. When comparing Figure. 2 B and D, we observe that the local minima found by SDT (Figure. 2 B) is not only sharper than that found by TET (Figure. 2 D) under $\mathcal{L}_{\text{SDT}}$ but also maintains a higher loss value. This supports our claim that TET loss cannot be easily minimized around sharp local minima (Figure. 2 B), thus preferable to converge into flatter local minima (Figure. 2 D). Put together, the results here provide evidence for our reasoning in Section 4.2.

**Training from SDT to TET.** In this part, we further validate the ability of TET to escape from the local minimum found by SDT. We adopt the VGGSNN with 300 epochs training on DVS-CIFAR10. First, we optimize $\mathcal{L}_{\text{SDT}}$ for 200 epochs and then change the loss function to $\mathcal{L}_{\text{TET}}$ after epoch 200. Figure 3 demonstrates the accuracy and loss change on the test set. After 200 epochs training, SDT

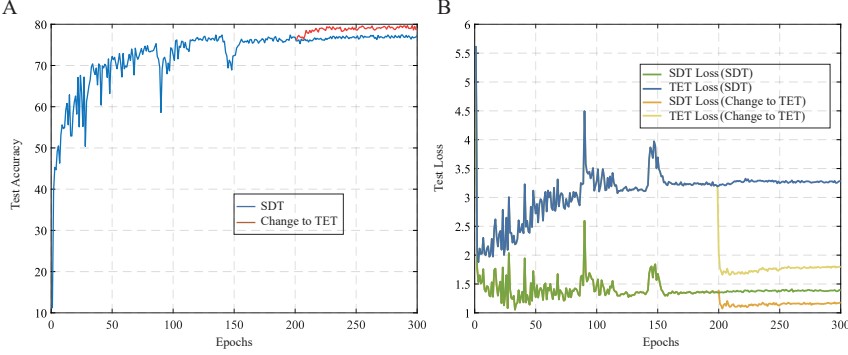

Figure 3: TET helps to jump out the local minimum point. We provide the test accuracy *(A)* and loss *(B)* change after changing the SDT to TET at epoch 200. TET efficiently improves the test performance and reduces the two kinds of loss.

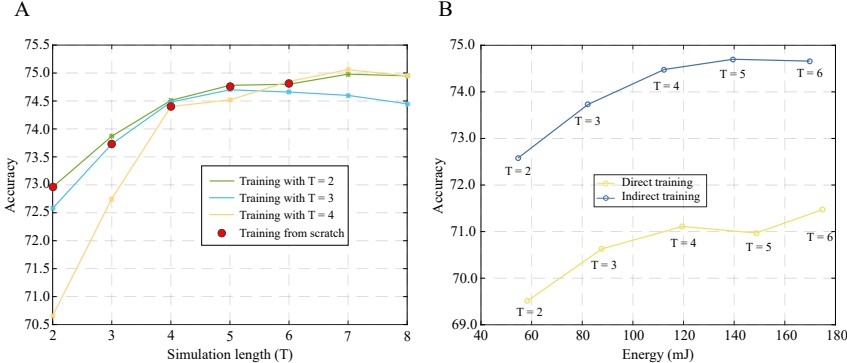

Figure 4: Time scalability robustness and network efficiency of ResNet-19 on CIFAR100. *(A)* The comparison of training from scratch (dots) and inheriting from a small simulation length (lines). *(B)* SNN network performance changes with energy consumption.

gets trapped into a local minimum, and the $\mathcal{L}_{\text{SDT}}$ no longer decreases. The $\mathcal{L}_{\text{TET}}$ is much higher than $\mathcal{L}_{\text{SDT}}$ since SDT does not optimize it. Nevertheless, after we change the loss function to $\mathcal{L}_{\text{TET}}$, the $\mathcal{L}_{\text{TET}}$ and $\mathcal{L}_{\text{SDT}}$ on the test set both have a rapid decline. This phenomenon illustrates the TET ability to help the SNN efficiently jump out of the local minimum with poor generalization and find another flatter local minimum.

**Time Scalability Robustness.** Here, we study the time scalability robustness of SNNs trained with TET ($\mathcal{L}_{\text{TET}}$). First, we use 300 epochs to train a small simulation length ResNet-19 on CIFAR100 as the initial SNN. Then, we directly change the simulation length from 2 to 8 without finetuning and report the network accuracy on the test set. Figure. 4. A displays the results after changing the simulation length. We use 2, 3, and 4, respectively, as the simulation length of the initial network. When we increase the simulation length, the accuracy of all networks gradually increases. After the simulation time reaches a certain value, the network performance will slightly decrease. Interestingly, SNNs trained from scratch (T=4 and T=6) are not as good as those trained following the TIT procedure.

**Network Efficiency.** In this section, we measure the relationship between energy consumption and network performance. SNN avoids multiplication on the inference since its binary activation and event-based operation. The addition operation in SNN costs $0.9pJ$ energy while multiplication operation consumes $4.6pJ$ measured in 45nm CMOS technology (Rathi & Roy, 2020). In our SNN model, the first layer has multiplication operations, while the other layers only have addition operations. Figure 4. B summarizes the results of different simulation times. In all cases, the SNN obtained by TET has higher efficiency.

## 5.2 COMPARISON TO EXITING WORKS

In this section, we compare our experimental results with previous works. We validate the full TIT algorithm ($\mathcal{L}_{\text{TOTAL}}$) both on the static dataset and neuromorphic dataset. All of the experiment results are summarized in Table 5.2. We specify all the training details in the appendix A.1.

**CIFAR.** We apply TET and TIT algorithm on CIFAR (Krizhevsky et al., 2009), and report the mean and standard deviation of 3 runs under different random seeds. The $\lambda$ is set to 0.05. On CIFAR10, our TET method achieves the highest accuracy above all existing approaches. Even when $T = 2$, there is a 1.82% increment compare to STBP-tdBN with simulation length $T = 6$. It is worth noting that our method is only 0.47% lower than the ANN performance. TET algorithm demonstrates a more excellent ability on CIFAR100. It has an accuracy increase greater than 3% on all report simulation lengths. In addition, when $T = 6$, the reported accuracy is only 0.63% lower than that of ANN. We can see that the proposed TET's improvement is even higher on complex data like CIFAR100, where the generalizability of the model distinguishes a lot among minima with different flatness.

Table 3: Compare with existing works. Our method improves network performance across all tasks. * denotes self-implementation results. † denotes data augmentation (Li et al., 2022).

| Dataset | Model | Methods | Architecture | Simulation Length | Accuracy |
|---|---|---|---|---|---|
| CIFAR10 | Rathi et al. (2019) | Hybrid training | ResNet-20 | 250 | 92.22 |
| | Rathi & Roy (2020) | Diet-SNN | ResNet-20 | 10 | 92.54 |
| | Wu et al. (2018) | STBP | CIFARNet | 12 | 89.83 |
| | Wu et al. (2019) | STBP NeuNorm | CIFARNet | 12 | 90.53 |
| | Zhang & Li (2020) | TSSL-BP | CIFARNet | 5 | 91.41 |
| | Zheng et al. (2021) | STBP-tdBN | ResNet-19 | 6 | 93.16 |
| | | | | 4 | 92.92 |
| | | | | 2 | 92.34 |
| | **our model** | TET | ResNet-19 | 6 | **94.50±0.07** |
| | | | | 4 | **94.44±0.08** |
| | | | | 2 | **94.16±0.03** |
| | ANN* | ANN | ResNet-19 | 1 | 94.97 |
| CIFAR100 | Rathi et al. (2019) | Hybrid training | VGG-11 | 125 | 67.87 |
| | Rathi & Roy (2020) | Diet-SNN | ResNet-20 | 5 | 64.07 |
| | Zheng et al. (2021)* | STBP-tdBN | ResNet-19 | 6 | 71.12±0.57 |
| | | | | 4 | 70.86±0.22 |
| | | | | 2 | 69.41±0.08 |
| | **our model** | TET | ResNet-19 | 6 | **74.72±0.28** |
| | | | | 4 | **74.47±0.15** |
| | | | | 2 | **72.87±0.10** |
| | ANN* | ANN | ResNet-19 | 1 | 75.35 |
| ImageNet | Rathi et al. (2019) | Hybrid training | ResNet-34 | 250 | 61.48 |
| | Sengupta et al. (2018) | SPIKE-NORM | ResNet-34 | 2500 | 69.96 |
| | Zheng et al. (2021) | STBP-tdBN | Spiking-ResNet-34 | 6 | 63.72 |
| | Fang et al. (2021) | SEW ResNet | SEW-ResNet-34 | 4 | 67.04 |
| | **our model** | TET | Spiking-ResNet-34 | 6 | **64.79** |
| | | TET | SEW-ResNet-34 | 4 | **68.00** |
| DVS-CIFAR10 | Zheng et al. (2021) | STBP-tdBN | ResNet-19 | 10 | 67.8 |
| | Kugele et al. (2020) | Streaming Rollout | DenseNet | 10 | 66.8 |
| | Wu et al. (2021) | Conv3D | LIAF-Net | 10 | 71.70 |
| | Wu et al. (2021) | LIAF | LIAF-Net | 10 | 70.40 |
| | **our model** | TET | VGGSNN | 10 | **77.33±0.21** |
| | | TET† | VGGSNN | 10 | **83.17±0.15** |

**ImageNet.** The training set of ImageNet (Krizhevsky et al., 2012) provides 1.28k training samples for each label. We choose the two most representative ResNet-34 to verify our algorithm on ImageNet with $\lambda = 0.001$. SEW-ResNet34 is not a typical SNN since it adopts the IF model and modifies the Residual structure. Although we only train our model for 120 epochs, the TET algorithm achieves a 1.07% increment on Spiking-ResNet-34 and a 0.96% increment on SEW-ResNet34.

**DVS-CIFAR10.** The neuromorphic datasets suffer much more noise than static datasets. Thus the well-trained SNN is easier to overfit on these datasets than static datasets. DVS-CIFAR10 (Li et al., 2017), which provides each label with 0.9k training samples, is the most challenging mainstream neuromorphic dataset. Recent works prefer to deal with this dataset by complex architectures, which are more susceptible to overfitting and do not result in very high accuracy. Here, we adopt VGGSNN on the DVS-CIFAR10 dataset, set $\lambda = 0.001$, and report the mean and standard deviation of 3 runs under different random seeds. Along with data augmentation methods, VGGSNN can achieve an accuracy of 77.4%. Then we apply the TET method to obtain a more generalizable optima. The accuracy rises to 83.17%. Our TET method outperforms existing state-of-the-art by 11.47% accuracy. Without data augmentation methods, VGGSNN obtains 73.3% accuracy by SDT and 77.3% accuracy by TET.

## 6 CONCLUSION

This paper focuses on the SNN generalization problem, which is described as the direct training SNN performs well on the training set but poor on the test set. We find this phenomenon is due to the incorrect SG that makes the SNN easily trapped into a local minimum with poor generalization. To solve this problem, we propose the temporal efficient training algorithm (TET). Extensive experiments verify that our proposed method consistently achieves better performance than the SDT process. Furthermore, TET significantly improves the time scalability robustness of SNN, which

enables us to propose the time inheritance training (TIT) to significantly reduce the training time consumption by almost a half.

## 7 ACKNOWLEDGMENT

This project is supported by NSFC 61876032 and JCYJ20210324140807019. Y. Li completed this work during his prior research assistantship in UESTC.

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

# A APPENDIX

## A.1 DATASET AND TRAINING DETAIL

**CIFAR.** The CIFAR dataset (Krizhevsky et al., 2009) consists of 50k training images and 10k testing images with the size of $32 \times 32$. We use ResNet-19 for both CIFAR10 and CIFAR100. Moreover, random horizontal flip and crop are applied to the training images the augmentation. First, we use 300 epoch to train the SNN with the simulation length $T = 2$. We use an Adam optimizer with a learning rate of 0.01 and cosine decay to 0. Next, following the TIT algorithm, we increase the simulation time (to 4 and 6) and continue training the SNN for only 50 epochs, with the learning rate changing to $1e - 4$.

**ImageNet.** ImageNet (Deng et al., 2009) contains more than 1250k training images and 50k validation images. We crop the images to 224×224 and using the standard augmentation for the training data. We use an SGD optimizer with 0.9 momentum and weight decay $4e - 5$. The learning rate is set to 0.1 and cosine decay to 0. We train the SEW-ResNet34 (Fang et al., 2021) with $T = 4$ for 120 epochs. As for the Spiking-ResNet34 (Zheng et al., 2021), we use TIT algorithm to train 90 epochs with $T = 4$ first, then change the simulation time to 6 and finetune the network for 30 epochs. We adopt an Adam optimizer on the finetune phase and change the learning rate to $1e - 4$. TIT algorithm significantly reduces the training time consumption since training the Spiking-ResNet34 is extremely slow.

**DVS-CIFAR10.** DVS-CIFAR10 (Li et al., 2017), the most challenging mainstream neuromorphic data set, is converted from CIFAR10. It has 10k images with the size 128×128. Following Samadzadeh et al. (2020), we divide the data stream into 10 blocks by time and accumulate the spikes in each block. Then, we split the dataset into 9k training images and 1k test images and reduce the spatial resolution to 48×48. Random horizontal flip and random roll within 5 pixels are taken as augmentation (Li et al., 2022). We adopt VGGSNN architecture with 300 epochs training on this classification task. And we use an Adam optimizer with the learning rate $1e - 3$ and cosine decay to 0. As for the case that does not apply any augmentation, we add a weight decay of 5e-4 to the optimizer.

## A.2 $\mathcal{L}_{SDT}$ LOSS LANDSCAPE OF RESNET-19

Here we compare the classification loss ($\mathcal{L}_{SDT}$) landscapes of ResNet-19 on CIFAR100. The position around the local minimal value found by the SDT ($\mathcal{L}_{SDT}$) is very sharp. However, the area around the local minimum found by TET ($\mathcal{L}_{TET}$) is much smoother (Figure 5), which indicates that TET effectively improves the network generalization. Such improvements could be further utilized to other techniques like privacy-preserving data generalization (Kim et al., 2021) and neural architecture search (Kim et al., 2022).

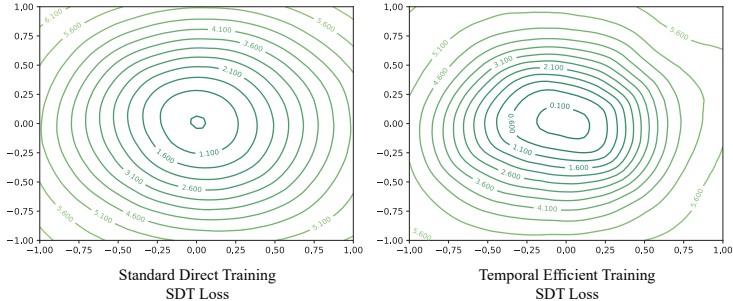

Figure 5: STD loss landscape of ResNet-19 on CIFAR100 from different training approaches.

## A.3 EFFECT OF $\mathcal{L}_{MSE}$

In this part, we examine the effect of the regular term $\mathcal{L}_{MSE}$ with 5 different levels of $\lambda$. Figure 6 Summarizes the final results. The regular term $\mathcal{L}_{MSE}$ effectively increases the performance of

both ResNet-19 on CIFAR100 and VGGSNN on DVS-CIFAR10. The static dataset CIFAR100 is more suitable for larger $\lambda$, while smaller $\lambda$ is suitable for DVS-CIFAR10. Theoretically, it is hard to obtain satisfying performance at the early simulation moment due to the sparseness of neuromorphic datasets. So too large regular term $\mathcal{L}_{\text{MSE}}$ is not suitable for the neuromorphic dataset. Furthermore, we find that a high $\lambda$ may harm the early training phase on ImageNet, especially if zero-initialize (Goyal et al., 2017) is not performed. As a result, we set $\lambda$ to $5e-2$ for CIFAR10 and CIFAR100, $1e-3$ for ImageNet and DVS-CIFAR10.

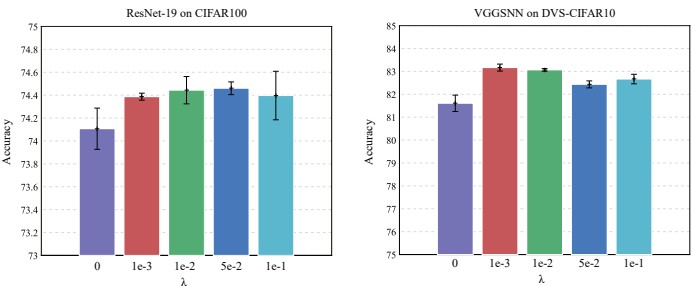

Figure 6: The accuracy under different levels of $\lambda$.

## A.4 STATISTICAL RESULTS

Here we provide statistical results (Figure 7) to prove that the total SNN accuracy is positively associated with every average of moment's output test accuracy. We train CNN-5 on CIFAR10 for a total of 20 runs with SDT and 5 runs with TET.

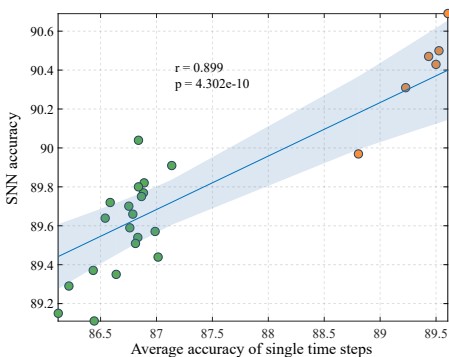

Figure 7: Statistical results. The overall performance of SNN is highly positively associated with the average accuracy of each moment. The standard training obtains the green dots, while the red dots are trained by the TET method.

## A.5 TIME SCALABILITY ROBUSTNESS OF SDT AND TET.

Here we first show the test accuracy (ResNet19 on CIFAR100) of the membrane potential increment at each moment instead of the integrated membrane potential. We set the initial simulation length of the SNNs to 3 or 4 and trained them for a full 300 epochs. Then we expand their simulation length to 8. As shown in table 4, TET ($\mathcal{L}_{\text{TET}}$) makes the membrane potential increment at each moment have a higher classification ability than SDT ($\mathcal{L}_{\text{SDT}}$). And TET (1.41 and 0.08) also acquires a low accuracy variance than SDT (3.81 and 4.04).

Then we compare the time scalability robustness between SDT ($\mathcal{L}_{\text{SDT}}$) and TET ($\mathcal{L}_{\text{TET}}$). We set the initial simulation length of ResNet19 SNNs to 2, 3, 4 and train with SDT or TET. Then we gradually increase SNN simulation length to 64 and record test accuracy of the integrated membrane potential.

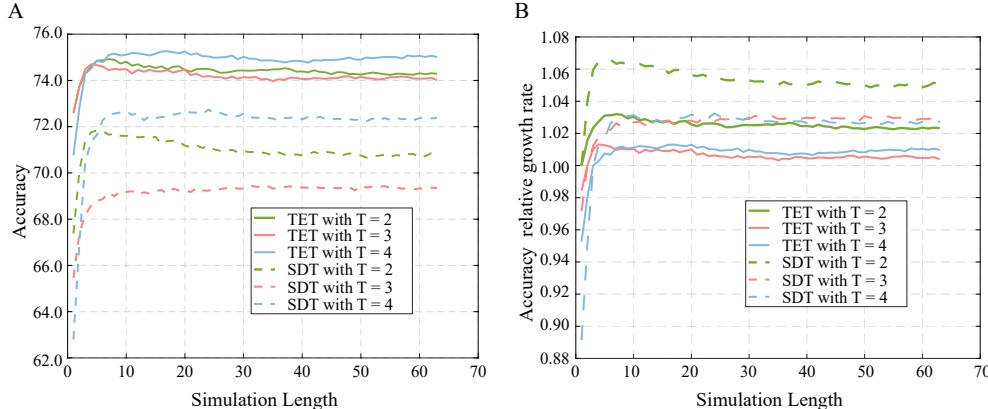

Figure 8: The accuracy after increasing the simulation length. We first train the SNN with TET (only use $\mathcal{L}_{\text{TET}}$) and SDT ($\mathcal{L}_{\text{SDT}}$) with simulation length (T) is 2, 3, or 4. Then, we increase the simulation to 64 without finetuning and record the test the classification accuracy (A) and the accuracy relative growth rate (B) of the total SNN output (integrate membrane potential) at each simulation time.

As we increase the simulation length, all the SNNs' accuracy will first increase and then be stable in a certain area. Meanwhile, TET (1.80) has a small accuracy variance than the SDT (11.13) after increasing the simulation length. This phenomenon indicates that the initialization steps of TIT only need a small simulation length SNN for TET but a sufficiently large simulation (or enough epochs for finetuning step) for SDT.

Table 4: Accuracy of each moment's membrane potential increment. We use $\mathcal{L}_{\text{SDT}}$ or $\mathcal{L}_{\text{TET}}$ to train the networks with simulation length 3 or 4. Then directly increase their simulation length to 8 and record each moment's potential increment test accuracy.

| Method | T=1 | T=2 | T=3 | T=4 | T=5 | T=6 | T=7 | T=8 |
|---|---|---|---|---|---|---|---|---|
| SDT (T=3) | 55.61 | 57.95 | 56.87 | 55.09 | 57.56 | 53.54 | 57.72 | 54.04 |
| SDT (T=4) | 37.96 | 61.78 | 55.03 | 56.64 | 57.47 | 54.24 | 58.74 | 55.48 |
| TET (T=3) | 65.97 | 72.22 | 71.78 | 70.55 | 71.90 | 69.57 | 72.15 | 69.78 |
| TET (T=4) | 62.17 | 71.57 | 71.05 | 72.08 | 71.77 | 71.23 | 71.81 | 71.36 |

