# OpenReview forum: "Temporal Efficient Training of Spiking Neural Network via Gradient Re-weighting"
_ICLR.cc/2022/Conference — ICLR 2022 Poster_

### Official Review · Reviewer_KVXN · 2021-10-20

**Correctness:** 3
**Technical Novelty And Significance:** 3
**Empirical Novelty And Significance:** 1
**Recommendation:** 8
**Confidence:** 4

**Main Review:**

There are some main issues to be addressed:
1. The TET method and the TIT method can be viewed as two individual parts. The TET method is a new temporal loss function and the TIT is a fine-tune method. If so, an ablation study is required to show the contribution of these two parts in temporal efficiency individually. Furthermore, can the TIT method be applied to other SNNs as long as these SNNs are trained with temporal loss (e.g., the STCA loss [1])?
2. In the section 4.3, this paper claims the TIT method could significantly reduce the training time. Here are some questions.
(1) the time spent per epoch is not linearly proportional to the simulation length, so it is inappropriate to claim ‘As a result, the TIT can reduce the training time cost by half.’
(2) based on the above analysis, actually, it seems difficult to select the training epoch and simulation length for the TIT method to achieve temporal efficiency for training SNNs.
3. In table 3, the network performance across static datasets and neuromorphic datasets, why the simulation length of two networks with the TET on the Imagenet are different? Is it from original paper or based on the optimization?
4. For time scalability, it’s better to supplement the result of relative growth rate for both SDT and TET.
5. Some minor comments, such as the unclear figure legends, the missing curve of “Training from scratch” in Fig 4A, inconsistent decimal places of accuracies and so on.

[1] Gu, et al. STCA: Spatio-Temporal Credit Assignment with Delayed Feedback in Deep Spiking Neural Networks. IJCAI 2019.



**Summary Of The Paper:**

This paper proposes a new training approach, the temporal efficient training (TET). This algorithm utilizes a new loss function to improve the generalizability of SNNs. Further, a new training pipeline is presented to reduce the simulation time of SNNs. This work outperforms the SOTA on the static datasets and neuromorphic datasets.

**Summary Of The Review:**

The novelty is limited, as the core idea and the relative derivations are straightforward.

---

> ### Author Response · Authors · 2021-11-16
> **Response to Reviewer KVXN**
>
> ### Response to Comment 1:
> Thanks for the comment on TET and TIT. The review is right in the sense that the two mechanisms can work separately but exists some misunderstanding on their utility. The TET method enables SNN to work with relatively short T for a high accuracy while TIT enables the training of SNN with high T to inherit from the trained model in low T.  In the current submission, we already perform the ablation study to demonstrate the validity of TET without TIT  in the first paragraph in Section 5.1.  For  TIT, its contribution to time efficiency is discussed in detail in Section 4.3. The response to  Review 2’s Comment 3 and Review 4’s Comment 2 (i.e. the next comment) also explained how TIT worked. In terms of the extension of using TIT on other methods, we expect TIT to work for STCA loss as well since  STCA shares the same training target as SDT and TIT. We test the time scalability robustness of STCA by changing the simulation length from 250 to 300 and 500, the accuracy of MNIST changes from 98.43% to 98.39% and 98.43%. This phenomenon shows that SNN trained with a small simulation length can be used as an initializer for SNN with larger T when the training loss is STCA, supporting the applicability of TIT in the training step.
>
> ### Response to Comment 2:
> As shown in the response to Review 2’s Comment 3, the training time and memory consumption are approximately linearly related to the SNN’s simulation time when the batch size is fixed. In terms of applying TIT to TET,  considering the fact that SNN trained with TET does not require a very high simulation length to converge, we can always start from T = 2 and increase T step by step. Suppose that the batch size is fixed over the TIT, based on the above analysis, the number of training epochs can be used as an estimation for the training cost.
>
> ### Response to Comment 3:
> In order to compare the current work with other existing works, we set the same simulation lengths as the original papers. We will make it more explicitly stated in the revision.
>
> ### Response to Comment 4:
> Thanks for your suggestion. We will add this complementary figure in the revision. In addition to comparing the whole SDT and TET trained models, we also provide the model accuracies solely on the membrane potential increment at each time point separately through TIT. It verifies that the TIT cannot dramatically improve the accuracy if the starting model in TIT is bad.  Please refer to our Response to Reviewer A3wt Part I Comment 2 for a related discussion.
>
> ### Response to Comment 5:
> "Training from scratch" represents the accuracy of the SNN with a large simulation length obtained from scratch training. This figure shows the effectiveness of TIT by illustrating the situation that increasing the simulation length of SNN (training with TET and a small simulation length) can obtain accuracy close to that when SNN is trained from scratch. We will fix the inconsistency issues in Fig 4A.

---

### Official Review · Reviewer_5F8z · 2021-10-31

**Correctness:** 3
**Technical Novelty And Significance:** 3
**Empirical Novelty And Significance:** 2
**Recommendation:** 5
**Confidence:** 4

**Main Review:**

Strength: The results are good compared to existing works. The paper also has a comprehensive comparison between the proposed TET and regular SDT.
Weakness: The proposed method only changes the loss function. The novelty may not be enough. It is still a kind of surrogate gradient method and cannot solve the approximation made by smoothing the gradient of the discontinuous spiking function.

**Summary Of The Paper:**

This paper proposed a new loss function called TET for directly training SNNs.

**Summary Of The Review:**

1. Missing reference: There’re some recent papers that should be discussed in the related works, e.g. [1]
2. I'd like to learn more about the details of data preprocessing. For example, does the real value in CIFAR and ImageNET datasets directly used as the inputs? In DVS-CIFAR10, how does the resolution reduced? How is the time length shrunk to 10 time steps?
3. Performance comparison: In CIFAR10, can the authors also provide the performance of the 5 layers CIFARNet for better comparison with existing works? In DVS-CIFAR10, do other works also use data augmentation, e.g.[2]? Does the network adopted in this paper have a similar number of parameters to the existing works in the comparison?



[1] Yang, Y., Zhang, W., & Li, P. (2021, July). Backpropagated Neighborhood Aggregation for Accurate Training of Spiking Neural Networks. In International Conference on Machine Learning (pp. 11852-11862). PMLR.
[2] Zhenzhi Wu, Hehui Zhang, Yihan Lin, Guoqi Li, Meng Wang, and Ye Tang. Liaf-net: Leaky integrate and analog fire network for lightweight and efficient spatiotemporal information processing. IEEE Transactions on Neural Networks and Learning Systems, 2021.

---

> ### Author Response · Authors · 2021-11-16
> **Response to Reviewer 5F8z**
>
> ### Response to Comment 1:
> Thanks for reminding us of this excellent paper. It provides a method that accelerates the SNN’s convergence and improves the performance. We will add it to the related work section.
>
> ### Response to Comment 2:
> We directly input the real value into the network for the static dataset (CIFAR and ImageNET) without using any encoding method. For the DVS-CIFAR10, We divide all events into ten segments according to event time and merge all events in each segment into one frame. We use the transforms.Resize function in PyTorch to reduce the resolution from 128 to 48, which means each point collects the information from the nearest points. We follow the same preprocessing pipeline as the reference paper [1]. Their preprocessing source code is provided on https://github.com/aa-samad/conv_snn.
>
> [1] Samadzadeh, Ali, et al. "Convolutional Spiking Neural Networks for Spatio-Temporal Feature Extraction." arXiv preprint arXiv:2003.12346 (2020).
>
> ### Response to Comment 3:
> For CIFARNet on CIFAR10 with simulation length T=4, the accuracy is 92.35% when training with SDT (the same training detail as ResNet-19), and increases to 92.86% when training with TET ($L_{TET}$).
>
> We find that the data augmentation has a significant effect on the DVS-CIAFR10 dataset. Without augmentation, the SDT only obtain 73.3% accuracy and the TET obtain 77.3% accuracy. Augmentation helps SDT to increase the accuracy to 77.4%. Moreover, continuing to change the SDT to TET will increase the accuracy to 83.1%. To address reviewers' concerns about data enhancement and parameter number, we use the LIAF-Net to verify our TET method. Since previous work did not share their code, we used our preprocessing pipeline (including data augmentation and resolution reduction) and built the LIAF-Net ourselves. We verify SDT and our TET results on the LIAF model, LIAF-Net achieve 70.5% accuracy by SDT and 74.3% accuracy by TET. Then we build the same network architecture but apply LIF model (with tdBN), we achieve 74.9% accuracy by SDT and 77.6% accuracy by TET.
>
> We will include the discussion on augmentation in the revision.

---

### Official Review · Reviewer_A3wt · 2021-11-02

**Correctness:** 1
**Technical Novelty And Significance:** 2
**Empirical Novelty And Significance:** 1
**Recommendation:** 5
**Confidence:** 5

**Main Review:**

$\textbf{Strengths:}$
The high classification accuracy on DVS-CIFAR10 is very impressive.

$\textbf{Weaknesses:}$
In spike of the excellent classification records achieved in this work, the authors’ work leaves a number of uncertainties that should be thoroughly addressed to not mislead the readers. My main concerns are as follows.

1.	I wonder if the use of TET is the main cause of the improvement in classification accuracy. The authors actually used $L_{TOTAL}=(1-\lambda)L_{TET}+\lambda L_{MSE}$ rather than $L_{TET}$. If the use of TET is the main cause, the comparable results should be achieved using solely $L_{TET}$. Given the use of $L_{TOTAL}$, the feature of TET such that loss evaluation at every timestep is undermined because $L_{MSE}$ is evaluated at the last timestep. Additionally, the loss landscape analysis is also undermined because the actually used loss is $L_{TOTAL}$ not $L_{TET}$. Therefore, the authors should clarify the effect of TET only if TET is the key to the present work.

2.	I wonder if the time inheritance training is the unique feature of TET. The authors should discuss if the same method can successfully apply to SDT as well. This method may be the main cause of the accuracy improvement. The authors should provide the general framework of the time inheritance training method to identify its scalability to different networks on different tasks rather than naively discussing about a single hypothetical case.

3.	The authors repeatedly remarked the mismatch of gradient and loss for SDT. I wonder on what ground the mismatch is assured and how the use of TET can resolve the mismatch. Also, the point that “TET needs the second term $\partial O/\partial W$ close to 0 to make the $L_{TET}$ convergence.” lacks its theoretical grounds, which should be proven. If this is true, the accuracy should be independent of timestep length T for training; yet, Figure 4 identifies that this is not the case. Even if it is true, this point cannot directly explain the accuracy improvement because learning was done with $L_{TOTAL}$

4.	There are many statements that may mislead the readers.
“Another training issue in the memory and time consumption …” on Page 1. Please specify when this is the truth.
“Furthermore, since the TET applies optimization on each time point…” on Page 2. Is this true for the use of $L_{TOTAL}$?
“TET algorithm optimizes each moment’s output, enabling us to extent…” on Page 6. I am very puzzled about this statement. What did the authors mean by “naturally extend the simulation time”?

5. The improvement in classification accuracy on DVS-CIFAR10 is impressive. However, for the moment, it is unclear the direct cause of the improvement for the reason that I remarked above.

Minor concerns:
There are many statements that may mislead the readers.
“Another training issue in the memory and time consumption …” on Page 1. Please specify when this is the truth.
“Furthermore, since the TET applies optimization on each time point…” on Page 2. Is this true for the use of $L_{TOTAL}$?
“TET algorithm optimizes each moment’s output, enabling us to extent…” on Page 6. I am very puzzled about this statement. What did the authors mean by “naturally extend the simulation time”?

**Summary Of The Paper:**

The authors propose the temporal efficient training (TET) method that features loss evaluation at every timestep. As such, the main difference between TET and SDT lies in the fact that TET evaluates loss at every timestep with the correct label vector whereas SDT evaluates loss at the last timestep. But unfortunately, this could not be done because of the use of $L_{TOTAL}=(1-\lambda)L_{TET}+\lambda L_{MSE}$, where $L_{MSE}$ is evaluated at the last moment. The proposed method is simple but its efficacy outperforms the SOTA results for various networks on various datasets.


**Summary Of The Review:**

Based on the main concerns listed above, I recommend for reject for the current form of manuscript.

---

> ### Author Response · Authors · 2021-11-16
> **Response to Reviewer A3wt Part I**
>
> ### Response to summary:
> The output layer of SNN will accumulate the membrane potential increment at each moment and finally output the mean value of total membrane potential as the classification indicator. The reviewer’s understanding of the difference between TET and SDT is correct that  $L_{TET}$ will calculate the classification loss of each moment's membrane potential increment and compute their average as the final training loss instead of using the classification loss of the mean value of the total membrane potential. However, there is some misunderstanding of the $L_{MSE}$ we used in the paper. The regularization loss  $L_{MSE}$ is used to avoid the outlier of membrane potential increment at each moment. As shown in Algorithm 1, we apply the $L_{MSE}$ loss at every $O(t)$ instead of evaluating it at the last timestep. In the inference phase, we use the same regularly used SNN inference rule that classifies based on the last layer’s average membrane potential for both TET and SDT.
>
> ### Response to Comment 1:
> Thanks for asking for clarification and hope our explanation can relieve the reviewer’s concerns here.  In terms of using L_Total, as we clarified above, it is calculated at each time rather than the last moment. In addition, it is not used in the results before Table 3.  The loss landscape analysis is based on $L_{TET}$ and $L_{SDT}$ instead of $L_{TOTAL}$ as we literally describe in the last paragraph on Page 6. Thus the improvement on the landscape is solely the effect of TET and has nothing to do with $L_{MSE}$. Regarding the $L_{MSE}$, since it can help stabilize the training, we include it as part of the loss when we construct the ultimate model to compare with others. Since it is not the central part, we included the discussion on $L_{MSE}$ in Appendix A.3 Effect of $L_{MSE}$. Putting the validation and ablation studies together, we are confident that the major improvement is from TET.
>
> ### Response to Comment 2:
> The time inheritance training (TIT) is not a unique feature of TET. It consists of two steps: training with a small simulation length and finetuning with a larger simulation length. It is used to save time by integrating the models with smaller simulation lengths into the models with larger simulation lengths. It won’t bring in any benefit in the accuracy when compared to training from scratch. Thus it would not be the main cause of the accuracy improvement. In terms of TIT for SDT, it works but requires more epochs for the finetuning to reach accuracy when training from scratch. The results are added as Appendix A5 in the revised manuscript. Moreover, TIT only applies to the static data set since the input at each moment has to be the same (images), and the neuromorphic data set is not suitable for TIT because the input at each moment is different.
>
> For ease of review, we attach the table of TIT on SDT v.s. TET solely on the membrane potential increment as well. It verifies that the TIT cannot dramatically improve the accuracy if the starting model in TIT is bad.
> #### ------------------------------------------------------------------------------------------------
> #### | Method      |  T=1   |  T=2  |   T=3  |   T=4  |   T=5  |  T=6  |   T=7  |  T=8  |
> #### | SDT (T=3) | 55.61 | 57.95 | 56.87 | 55.09 | 57.56 | 53.54 | 57.72 | 54.04|
> #### | SDT (T=4) | 37.96 | 61.78 | 55.03 | 56.64 | 57.47 | 54.24 | 58.74 | 55.48|
> #### | TET (T=3) | 65.97 | 72.22 | 71.78 | 70.55 | 71.90 | 69.57 | 72.15 | 69.78 |
> #### | TET (T=4) | 62.17 | 71.57 | 71.05 | 72.08 | 71.77 | 71.23 | 71.81 | 71.36 |
> **Accuracy of each moment's membrane potential increment.** We use $\mathcal{L}_\text{SDT}$ or $\mathcal{L}_\text{TET}$ to train the networks with simulation length 3 or 4. Then directly increase their simulation length to 8 and record each moment's potential increment test accuracy.
> #### ------------------------------------------------------------------------------------------------

---

> > ### Comment · Reviewer_A3wt · 2021-11-19
> > **Comments on responses**
> >
> > I agree with Authors that I misunderstood $L_{MSE}$. This is because the term $L_{MSE}$ is not explained anywhere in the manuscript.
> > Comment on Response to Comment 1:  To be honest, I do not get the point of the loss analysis of $L_{TET}$. The performance breakthrough was achieved by using $L_{TOTAL}$. But, the authors attempted to figure out the reason behind the breakthrough from $L_{TET}$ analysis rather than $L_{TOTAL}$. Do I miss something?

---

> > > ### Author Response · Authors · 2021-11-19
> > > **Clarification on the Loss Contribution to the Ultimate Model**
> > >
> > > Thanks for pointing out  the lack of explanation of the $L_{MSE}$. We will add it to the new manuscript.
> > >
> > > In terms of the loss analysis, it is true that we use the $L_{Total}$ for the overall comparison. However, we also provide ablation studies to demonstrate how each part contributes. In our experiments, most of the performance breakthrough comes from $L_{TET}$ rather than $L_{MSE}$. As shown in Fig 6, compared to $L_{TET}$ ($\lambda=0$), $L_{Total}$ ($\lambda>0$)  continues to improve the accuracy by 0.34% on CIFAR100 and 1.4% on DVS-CIFAR. These improvements are much less than the improvements from $L_{MSE}$ to $L_{TET}$ (3.5% on CIFAR100 and 4.3% on DVS-CIFAR). This is why we focus on $L_{TET}$ for more detailed analysis in the current paper. Maybe we would discuss the effect of regularization like $L_{MSE}$ in the future.

---

> ### Author Response · Authors · 2021-11-16
> **Response to Reviewer A3wt Part II**
>
> ### Response to Comment 3:
> Since the SNN model contains the non-differentiable step function, it commonly uses a surrogate gradient to approximate the derivative. However, the surrogate gradient is inaccurate, making the network training by SDT easily trapped into local minimal instead of global minimal. The TET did not resolve the mismatch. Instead, the TET relieves the impact of the mismatch by guiding the network to converge to flat local minimal and obtain a higher test accuracy (as we showed in the comparison of loss landscape). Typically as T increases, the expression ability of the network will increase, which leads to an increment of accuracy. So we are not sure about the saying "If this is true, the accuracy should be independent of timestep length T for training." In addition, to prove that TET makes the second term $\frac{\partial O}{\partial W}$ close to 0, we compute the resnet19 (T=6, CIFAR100)   $\frac{\partial O}{\partial W}$  L2 norm and calculate the sum L2 norm that means the summation of all the layers. As a result, we find the sum L2 norm decrease from 0.7303 (SDT) to 0.0127 (TET). We agree that it would be more supportive if we can mathematically prove the efficacy, but admittedly we haven't figured out the proof right now and can only demonstrate the efficacy with ablation studies to eliminate the other possibilities.
>
> ### Response to Comment 4:
> “Another training issue in the memory and time consumption …” To specify the truth of this sentence, we train a ResNet19 on CIFAR-100 on four TITAN Xp GPUs, and the batch size is 128. We change the simulation length to 1,2,3,4 and test the average training time and memory consumption for an epoch. The following table shows the required average epoch time consumption and memory per GPU as the simulation time increase.  Both the time and memory consumption are approximately linearly related to simulation time. The sentence “Furthermore, since the TET applies optimization on each time point…” is true when using the $L_{TOTAL}$ since the loss $L_{MSE}$ is also calculated at each moment. TET optimizes each moment's output (membrane potential increment), the SNN has a higher generalization ability, and every moment's network input is the same. So the increment of next time also has a high classification accuracy and promotes the performance of the overall membrane potential to a certain extent. The statement "naturally extend the simulation time." is based on the experiments of Figure 4 A that as we directly increase the simulation length without any finetune, the SNN accuracy will increase to near the accuracy of training from scratch.
>  ```
>                              |     T=1     |      T=2   |    T=3        |      T=4
> Training time per epoch     |   68.23s    |   135.54s  |   217.91s     |    295.41s
> Training memory per GPU     |  1413MB     |   2433MB   |   3367MB      |    4355MB
>
>
> ```
> ### Response to Comment 5:
> We think the reviewer’s concern is mainly from Comments 1 & 2. As we have already clarified above, the MSE Loss and TIT are definitely not the cause of improvement. Although currently, we are unable to theoretically prove the efficacy of TET, our sufficient ablation study has excluded the alternative possibility.

---

> > ### Comment · Reviewer_A3wt · 2021-11-19
> > **Comments on responses**
> >
> > Regarding the authors' response to comment 3, it is hardly convincing that TET can drive the parameters toward flat global minimum. A few loss landscape on a few data cannot be evidence for the authors' bold assertion.
> >
> > Regarding the response to comment 4, I would like to ask the authors to take a careful look at the theoretical time and space complexity of the algorithm. I do not find any reasons for the exponential growth of the complexities with timestep. The authors' measures of the complexities appears rather linear than exponential.

---

> > > ### Author Response · Authors · 2021-11-19
> > > **Response to Reviewer A3wt's Further Comments**
> > >
> > > Thanks for your further comments.
> > >
> > > We agree that a mathematically proof of the $L_{TET}$ efficacy will be more convincing, but it is very tough to achieve. This is why we used the landscape as an alternative. In terms of the loss landscape, it is calculated on the whole training set rather than a few data. Thus it is representative for the model on the whole dataset. In addition, we demonstrated it for both CIFAR-100 and DVS-CIFAR and observed the same phenomenon. Due to the page limit, the loss landscape plot for CIFAR-100 was attached in Appendix A2. As complementary support, we also calculated the largest eigenvalue ($\lambda_{max}(Hess)$) and the trace ($Tr(Hess)$)of the Hessian at the found local minima for TET and SDT. For ResNet 19 on CIFAR 100 with T=4 and 128 images, the SDT showed $\lambda_{max}(Hess)=2.71$, $Tr(Hess)=125.42$ while the TET showed $\lambda_{max}(Hess)=0.24$, $Tr(Hess)=1.11$. These quantities also support that the minima discovered by TET are flatter than that of SDT. For the theoretical concern, given $L_{SDT}\leq L_{TET}$, the local minimum for $L_{TET}$ is also the local minimum for $L_{SDT}$. It then suffices to show that $Hess(L_{SDT};W\in\arg\min L_{SDT}\backslash\arg\min L_{TET})\succeq Hess(L_{SDT};W\in\arg\min L_{TET})$. We tried to prove this by plugging in the gradient condition and decomposing the Hessian into two semidefinite parts $E_{x}\left[\frac{\partial O}{\partial W}\frac{\partial O}{\partial W}^T\right]$ and $E_{x}\left[\frac{\partial ^2 O}{\partial W^2}\right]$, but haven't yet figured out the details. We think that our current empirical results support the assertion that the TET can help drive the parameters toward a flatter minimum, although why it works in such a way may need future exploration both theoretically and experimentally.
> > >
> > > Thanks for pointing our mistake in the statement here. You are right that the time and space complexity is linearly associated with the simulation length rather than exponential. We will correct the corresponding words in the manuscript.

---

### Official Review · Reviewer_HqdL · 2021-11-05

**Correctness:** 3
**Technical Novelty And Significance:** 3
**Empirical Novelty And Significance:** 3
**Recommendation:** 8
**Confidence:** 4

**Main Review:**

While the proposed improvements, TET and TIT, may appear only as minor tweaks as compared to previous work, the demonstrated improvements in speed and accuracy suggest that the paper may have substantial impact for SNN research community.

**Summary Of The Paper:**

The paper focusses on an important problem of improving supervised training of spiking neural nets (SNNs), as the current state of SNN training is lacking as compared to that for standard ANNs. The authors propose a couple simple but effective methods for improving both training efficiency and classification accuracy:  temporal efficient training (TET) which simply accumulates error gradients for each time step, and time inheritance training (TIT), which is a repeated training schedule with increasing simulation times.  In extensive experiments the authors demonstrate that their approach improves training time and accuracy.

**Summary Of The Review:**

Efficient supervised learning of deep SNNs is a problem that has resisted research efforts for many decades, so even incremental improvements are welcome in this area. The results of the paper offer a much desired step towards better SNN backprop algorithms that promise to unleash the many advantages of SNNs.
The paper is lucidly written and provided experiments strongly support the claims.

---

### Public Comment · ~Junkyu_Kim1 · 2021-11-13
**Concerns about the "spiking neuron" and its corresponding advantages.**

I want to know what is the essential difference between this network and artificial neural networks? Why can this network be claimed as the spiking neuron network? Can I consider the "spiking neuron" in this paper as an Activation Function different from ReLU?

I would appreciate it if you could answer my questions.

---

> ### Author Response · Authors · 2021-11-16
> **Difference between SNN and ANN**
>
> Thanks for your interest in our work. Spiking neurons differentiate from regular neurons in the way it deals with the integrated input. For example, suppose the integrated input is $Wx+b$ on some layer. The input to ANN is instant but the input to SNN is a sequence. The regular ANN neuron would send $\sigma(Wx+b)$ to the next layer with some activation function $\sigma(\cdot)$, while the SNN neuron would send $V_{th}$ to the next layer if $Wx+b \geq V_{th}$ and 0 otherwise, and keep the residual in memory waiting for the input at next moment.
>
> Under this setup, we can see that SNN eliminates the multiplication by always emitting a fixed value of output, which theoretically leads to computational efficiency if the hardware supports the deployment, e.g. Loihi chips from Intel. Another benefit is its adaptivity to deal with eventual input like the DVS dataset we used in the current paper. Other benefits are under exploration as well.
>
> In terms of mimicking the brain network, there are spectrums of models varying in their biological honesty and computational efficiency. The Integrate-and-Fire model we adopt here is the simplest in biology and most efficient for computation. There exist more complex models like the Leaky integrate-and-fire model, spiking response model, Hodgkin–Huxley model, etc.

---

> > ### Public Comment · ~Junkyu_Kim1 · 2021-11-17
> > **Not Convincing**
> >
> > Thanks for your response. But I am still concerned about this question.
> >
> > (1) As you illustrated, the activation of a spiking neuron is $\mathcal{F}(Wx+b)$, where $\mathcal{F}=V_{th}\ ({\rm if}\ Wx+b\ge V_{th})$ or $\mathcal{F}=0\ ({\rm if}\ Wx+b< V_{th})$. This function is extreamly similar to the $\rm step(\cdot)$ function with a scale transform of $V_{th}$, as represented as $V_{th}\ \rm step(\cdot)$. However, the $\rm step(\cdot)$ activation function does not perform satisfactorily in deep neural networks. Thus, I am really suspicious of the performance of your models.
> >
> > (2) As you reported, you achieved over 10\% improvement compared to existing SOTA works. However, you do not release the code. Could you share your training scripts and logs? I think open source will make your work being more convincing.
> >
> > (3) As you claimed that "SNN eliminates the multiplication", the membrane potential of your spiking model is calculated by $Wx+b$. How to eliminate the multiplication of $\bf{Wx}$
> >
> > (4) As you claimed that "it leads to theoretically computational efficiency when deployed on Loihi". Is it possible to provide the corresponding theoretical calculation?
> >
> > (5) You claimed that the Integrate-and-Fire model you adopted is the simplest in biology. Can your algorithm be generalized to more complex neuron models (e.g., Leaky integrate-and-fire model, spiking response model, Hodgkin–Huxley model)? If not, could you please clarify the limitation?
> >
> > An interesting work but existing several confusing problems, I think.

---

> > > ### Author Response · Authors · 2021-11-18
> > > **For your questions**
> > >
> > > Thanks for your further interest and questions.
> > > (1) You are right that the step function cannot work satisfactorily for DNN directly due to its discontinuity. But it does not mean it cannot work after modification. For example, you can also refer to the benchmarks we compared in the paper. Many of them actually proposed different approaches to deal with continuity. In this work, we also provide a different approach to deal with the impact of discontinuity to improve the performance of SNN.
> > >
> > > (2) We are confident about the reproducibility as we already performed the experiments with different seeds many times by ourselves. We will definitely release the code after the review with training scripts and logs, especially on the CIFAR-DVS one you mentioned.
> > >
> > > (3) The $x$ here is actually the output from the previous layer thus a discretized value in  $V_{th}$ and 0. So $Wx$ would be a summation rather than multiplication. In addition, through scaling, you can also set $V_{th}=1$. Thus the computation of $Wx$ can be highly simplified in the implementation.
> > >
> > > (4) You can refer to [1] for a practical application of SNN with Loihi and [2] for the details of estimating the computational cost.
> > >
> > > (5) Most algorithms are model-dependent. We won't overclaim that our model can work for more complex neuron models without performing the related experiments. Indeed, how to build a computational-efficient model with complex neuron dynamics is a quite challenging problem for the whole machining learning and computational neuroscience groups. It is an open problem rather than a special limitation for the current work.
> > >
> > > Thank you again for your interest in the SNN topic and hope our explanation can help you understand the field.
> > >
> > > ------------------------------------------------------------------------------------------------------------------------------
> > >
> > >
> > > [1] Massa, Riccardo, et al. "An efficient spiking neural network for recognizing gestures with a dvs camera on the loihi neuromorphic processor." arXiv preprint arXiv:2006.09985 (2020).
> > >
> > > [2] Rathi, Nitin, and Kaushik Roy. "DIET-SNN: Direct input encoding with leakage and threshold optimization in deep spiking neural networks." arXiv preprint arXiv:2008.03658 (2020).

---

> > > > ### Public Comment · ~Junkyu_Kim1 · 2021-11-18
> > > > **Your answers seem not to solve the problems.**
> > > >
> > > > Thank you for your patient response. But you didn't address all of the questions.
> > > >
> > > > (1) As you mentioned in your paper that “$\textbf{we regard the SNN as RNN and calculate the gradients through spatial-temporal backpropagation}$” and you responded above that "we also provide a different approach to deal with the impact of discontinuity " for the $\rm step(\cdot)$ function, both your neuron model and training algorithm is a modification of ANNs, why can you determine that your model is more efficient than ANN models when deployed on neuromorphic hardware?
> > > >
> > > > (2) You give the reference of [1] and [2]. However, [1] adopts the ANN-SNN conversion method, which is different from your directed training method. $\textbf{[1] and yours are two different technical routes.}$ Why are you sure that your method will work on Loihi as well? Besides, [2] are not really deployed on Loihi. Even if they claim the energy-efficiency advantage, it does not mean that your approach has the advantage as well. Because your method is different from [2].
> > > >
> > > > (3) You claimed that the $V_{th}$ can be set to 1 in your model. However, I am concerned that whether your proposed algorithm can work in this setting? Experimentally validated? Even though it can work, Is your method still comparable to existing works after the performance degradation caused by setting $V_{th}=1$?
> > > >
> > > > (4) Your weights are not binarized, which means that $W$ are floating points. $\textbf{When $V_{th}\neq 1$, the $Wx$ is not possible to be converted to pure summation operations.}$. Thus, compared to ANNs, your approach has no advantage in computational efficiency.

---

> > > > > ### Author Response · Authors · 2021-11-18
> > > > > **Our last response to your questions about general setups in SNN**
> > > > >
> > > > > (1) and (2) First, I think you may misunderstand SNN network efficiency, which means SNN is more efficient in the inference phase than ANN rather than in the training phase.  Direct training (usually less than ten) can obtain an SNN with a smaller simulation length than ANN-to-SNN conversion (usually tens of hundreds). Our models (Eqn 1-3), [1], and [2] both follow the IF or LIF model, and each layer’s forward process of the three works is the same. So no matter how to obtain the SNN models, the inference phase will be the same when the SNN models have the same architecture, parameters, and simulation length.
> > > > >
> > > > > (3) As we mentioned in the explanation of Eqn 2-3, the Vth is set to 1 for all of our experiments. The direct training method also works even if we train SNN after changing the threshold value and fixing the spike to 1 (final accuracy may be changed).
> > > > >
> > > > > (4) When the spike is equal to the threshold and every layer has the same threshold, you can always change the spike and threshold to 1 without any information loss (this step is follow weight normalization of ANN-to-SNN conversion [3] part 5).
> > > > >
> > > > > [1] Zheng, Hanle, Yujie Wu, Lei Deng, Yifan Hu, and Guoqi Li. "Going deeper with directly-trained larger spiking neural networks." arXiv preprint arXiv:2011.05280 (2020).
> > > > > [2] Fang, Wei, Zhaofei Yu, Yanqi Chen, Tiejun Huang, Timothée Masquelier, and Yonghong Tian. "Deep residual learning in spiking neural networks." In Thirty-Fifth Conference on Neural Information Processing Systems. 2021.
> > > > > [3] Sengupta A, Ye Y, Wang R, et al. Going deeper in spiking neural networks: VGG and residual architectures[J]. Frontiers in neuroscience, 2019, 13: 95.

---

### Author Response · Authors · 2021-11-16
**General Response to All Reviews and Comments**

We really appreciate the effort of all the reviewers for their insightful comments to help improve the quality of the current work. We will revise accordingly in the next-version manuscript. However, there are also some misunderstandings of our work that may affect the evaluation of the current submission. So we want to make a general response first to clarify our contribution and hope it could relieve the reviewers’ concerns.


First, we want to emphasize that our contribution is more than simply the change of the loss although the proposed TET is quite efficient in practice. We believe that it is more valuable that we perform an intensive investigation on the impact of adopting surrogate gradient in SNN training, from which we shed light on how to understand the model generalization of SNN beyond its biological analog.


Second, in terms of ablation study, we DID NOT use  $L_{MSE}$  in Figure 2 to validate the efficacy of TET over SDT. The $L_{MSE}$ was only used in the models presented in Table 3 for the purpose of regularization. Regarding TIT, it is a technique to accelerate training rather than improve accuracy. It can be well adapted to other existing methods as well.

We also respond to each reviewer with additional results. Please refer to our response to each comment for more technical details.

-------------------------------

As a response to the concerns about the reproducibility of our work, we pre-release the code for training SNN on DVS-CIFAR in the supplementary material. However, due to the time limit, we are not able to add enough comments to make the codes readable enough. But we believe that it can be used for reproducibility. We will release a cleaner version later.

---

> ### Public Comment · ~Junkyu_Kim1 · 2021-11-18
> **More concerns about the contribution and benefits of the paper.**
>
> （1）$\textbf{What is the advantage of your model against binary neural networks (BNNs)?}$ For BNNs, both the weights and activation can be compressed to 1-bit [1]. For your model, the weights are not binarized. The activation function you claimed can be binarized (0/1), but actually, you do not do that and do not perform corresponding experiments. Moreover, both the training and inference of BNNs only need one time-step rather than the multiple time steps by your model.
>
> （2）$\textbf{The main contribution is a loss function (i.e., Eq.9 in paper).}$ You have experimentally verified the boost (very impressive improvements) from this loss, but you don't release the code and training details. In addition, you do not illustrate clearly through theory why it leads to the improvement. This is what keeps me confused!
>
> [1] Qin et al. "Binary neural networks: A survey." Pattern Recognition. 2020

---

> > ### Author Response · Authors · 2021-11-18
> > **Our last response to your questions.**
> >
> > Please read our response to the other reviewers carefully.  Also, in terms of the general questions for a general spiking neural network, it is neither possible for us to explain every word from the beginning in a paper nor efficient to address every common question in one paper. You may easily access the literature online through google scholar if you are really interested in this topic.
> >
> > In terms of the code, as we respond below, we will release our code later as we did before in our other papers. We cannot point those links to you right now due to the anonymous requirement during this review period.
> >
> > While we are open to discussing, we don't accept the irresponsible blames without reasoning and details.

---

> > > ### Public Comment · ~Junkyu_Kim1 · 2021-11-18
> > > **Last Comment: Concerns on REPRODUCIBILITY**
> > >
> > > I am also a researcher in neuromorphic computing. I don't understand, even if you don't want to illustrate those problems existing in your paper clearly, why do you treat me as an outsider?
> > >
> > > A few days before, I was glad that your method achieved surprising results, which seems to have the potential to promote the SNNs. Consequently, $\textbf{I reproduced your method in codes, but did not achieve the performance you reported in your paper}$ (actually, the effectiveness shown in my experiments is very limited). So, it is very confusing to me... Maybe there is something wrong with my reproduced code. But I think the probability of existing errors is very little. Because the main contribution of your work is just a loss function compared with previous works, easy to be implemented.
> > >
> > > $\bigstar$ Releasing the code on this website does not violate the anonymity rule. After all, many ICLR-2022 papers provide the code in an attachment or via a link...
> > >
> > > This is also my last comment on this paper. Since you treat academics in such an attitude, I will never continue to follow your works at all.
> > >
> > > $\textbf{I DECLARE:}$ I am not attacking you much less blaming you. I don't know your identity under anonymity. I am just taking academic issues seriously.

---

> > > > ### Public Comment · ~Wei_Fang2 · 2021-11-19
> > > > **Experiments about temporal efficient training cross entropy**
> > > >
> > > > Hi, I noticed this paper before. The temporal efficient training cross entropy (TET CE) is an interesting method and easy to implement. I tested it in a simple network on CIFAR10-DVS and DVS Gesture. The results show that TET CE loss can promote test accuracy (CIFAR10-DVS: 66.6->70.1, and DVS Gesture: 79.17 -> 80.9). Note that I do not aim to reproduce the results in this paper with large-scale networks. I just check if this loss helps. I used a shallow network and trained it with only 16 epochs.
> > > >
> > > > As the TET CE loss is helpful, I added it to our SpikingJelly framework two weeks ago at this [commit](https://github.com/fangwei123456/spikingjelly/commit/5cd4cf28a91b75e801411e4008495d87cc5964fb). The API doc is  [spikingjelly.clock_driven.functional.temporal_efficient_training_cross_entropy](https://spikingjelly.readthedocs.io/zh_CN/latest/spikingjelly.clock_driven.functional.html#spikingjelly.clock_driven.functional.temporal_efficient_training_cross_entropy).
> > > >
> > > > Here are the model codes:
> > > >
> > > > ```python
> > > > import torch
> > > > import torch.nn as nn
> > > > import torch.nn.functional as F
> > > > from spikingjelly.clock_driven import functional, neuron, surrogate, base, layer
> > > >
> > > > class Conv3x3(nn.Module):
> > > >     def __init__(self, in_channels, out_channels, multi_step_neuron: callable = None, **kwargs):
> > > >         super().__init__()
> > > >         self.conv = nn.Conv2d(in_channels, out_channels, kernel_size=3, padding=1, bias=False)
> > > >         self.bn = nn.BatchNorm2d(out_channels)
> > > >         self.sn = multi_step_neuron(**kwargs)
> > > >
> > > >     def forward(self, x_seq):
> > > >         x_seq = functional.seq_to_ann_forward(x_seq, [self.conv, self.bn])
> > > >         return self.sn(x_seq)
> > > >
> > > >
> > > > class SEWBlock(nn.Module):
> > > >     def __init__(self, cnf, num_conv, channels, multi_step_neuron: callable = None, **kwargs):
> > > >         super().__init__()
> > > >         self.cnf = cnf
> > > >         conv = []
> > > >         for _ in range(num_conv):
> > > >             conv.append(Conv3x3(channels, channels, multi_step_neuron, **kwargs))
> > > >         self.conv = nn.Sequential(*conv)
> > > >
> > > >
> > > >     def extra_repr(self) -> str:
> > > >         return super().extra_repr() + f'cnf={self.cnf}'
> > > >
> > > >     def forward(self, x_seq: torch.Tensor):
> > > >         identity = x_seq
> > > >         if self.cnf == 'IAND':
> > > >             x_seq = self.conv(x_seq)
> > > >             return identity * (1. - x_seq)
> > > >         elif self.cnf == 'IANDINV':
> > > >             x_seq = self.conv(1. - x_seq)
> > > >             return identity * (1. - x_seq)
> > > >
> > > > class SEWResNet(nn.Module):
> > > >     def __init__(self, cnf, down_num: int, channels: int, blocks_num: int, conv_per_block=2):
> > > >         super().__init__()
> > > >         conv = []
> > > >         in_channels = 2
> > > >         out_channels = channels
> > > >         for _ in range(down_num):
> > > >             conv.append(Conv3x3(in_channels, out_channels, neuron.MultiStepParametricLIFNode, surrogate_function=surrogate.ATan(), detach_reset=True, backend='cupy'))
> > > >             for i in range(blocks_num):
> > > >                 conv.append(SEWBlock(cnf, conv_per_block, out_channels, neuron.MultiStepParametricLIFNode, surrogate_function=surrogate.ATan(), detach_reset=True, backend='cupy'))
> > > >             conv.append(
> > > >                 layer.SeqToANNContainer(nn.MaxPool2d(2, 2))
> > > >             )
> > > >             in_channels = out_channels
> > > >
> > > >         self.conv = nn.Sequential(*conv)
> > > >
> > > >         w = h = 128 >> down_num
> > > >
> > > >         self.fc = nn.Linear(in_channels * w * h, in_channels)
> > > >         self.sn = neuron.MultiStepParametricLIFNode(surrogate_function=surrogate.ATan(), detach_reset=True, backend='cupy')
> > > >         self.fc2 = nn.Linear(in_channels, 10)
> > > >
> > > >     def forward(self, x_seq):
> > > >         x_seq = x_seq.float().permute(1, 0, 2, 3, 4)  # [T, N, 2, *, *]
> > > >         x_seq = self.conv(x_seq).flatten(2)
> > > >         x_seq = functional.seq_to_ann_forward(x_seq, self.fc)
> > > >         x_seq = self.sn(x_seq)
> > > >         x_seq = self.fc2(x_seq.mean(0))
> > > >
> > > >         return x_seq
> > > >  class SEWResNet2(SEWResNet):
> > > >     def forward(self, x_seq):
> > > >         x_seq = x_seq.float().permute(1, 0, 2, 3, 4)  # [T, N, 2, *, *]
> > > >         x_seq = self.conv(x_seq).flatten(2)
> > > >         x_seq = functional.seq_to_ann_forward(x_seq, self.fc)
> > > >         x_seq = self.sn(x_seq)
> > > >         x_seq = functional.seq_to_ann_forward(x_seq, self.fc2)
> > > >
> > > >         return x_seq
> > > > ```
> > > >
> > > > The hyper parameter is:
> > > >
> > > > ```shell
> > > > -channels 128 -down_num 5 -blocks_num 1 -conv_per_block 2 -epochs 16 -lr 0.1 -T 4 -cnf IANDINV
> > > > ```

---

> > > > > ### Author Response · Authors · 2021-11-29
> > > > > **Thanks for providing the reproducible codes**
> > > > >
> > > > > Thanks for providing the codes for replicating our proposed method to support us. We will also release our own version soon.

---

> > > > ### Author Response · Authors · 2021-11-19
> > > > **Clarifications**
> > > >
> > > > Hi,
> > > >
> > > > First of all, we thank you for your interest in this work. I will give clarifications to your questions.
> > > >
> > > > 1. Difference between SNN and BNN? Reply: there are lots of differences between them. SNN deals with binary activation (and of course full precision weights), it also leverages $T$ Time steps temporal information. So you can see our SNN has higher accuracy on CIFAR10. In terms of hardware, SNN and BNN are not the same. BNN is usually used in Von-Neumann architecture hardware, while SNN is applied to neuromorphic hardware, where 0 activations (no spike) won’t involve computation. In fact, I am an expert in quantization and I know authors from Qin et al. 2020. Explaining the difference between SNN and BNN is beyond our paper’s scope. I would suggest you read more fundamental SNN papers. If you want, I can give you recommendations.
> > > >
> > > > 2. Reproducibility issue. Reply: We know that submitting codes to this website does not violate anonymity rules. But we prefer open-source code if the paper can be accepted, which is what we do previously. To address your concern, we provide the code of DVS-CIFAR10 in supplemental. Also, Wei Fang provides further experiments (even two weeks ago, as he commented). We’d like to thank him for his interest in our work.
> > > >
> > > > Here are our questions and we hope you can reply:
> > > >
> > > > 1. As a researcher in neuromorphic computing, why you didn’t know much about spiking neural networks and neuromorphic hardware?
> > > >
> > > > 2. May I know the details and results of your experiments? By commenting “*the effectiveness shown in my experiments is very limited*”, we would like to know what is your implementation model, dataset, and training hyper-parameters? Could you provide code like Fang Wei does?
> > > >
> > > > 3. As a researcher at Google, why you did not open public your research and education experience as well as the homepage, considering you are taking academic issues seriously.

---

### Author Response · Authors · 2021-11-29
**Would you please re-evaluate our paper after rebuttal?**

Dear reviewers and AC/PC chairs,

We really appreciate all the insightful comments on our work and sincerely hope that you can re-evaluate our work after the rebuttal during which we have provided more evidence and clarification to verify the proposed method.

Admittedly we did not mathematically prove the efficacy of the proposed method considering the difficulty in generalization analyses even for a simplified ANN model, but we have performed abundant ablation studies and comparative experiments to demonstrate its superiority over existing methods in model accuracy. Moreover, we provided a heuristic explanation from the loss landscape to illustrate why it generalizes better from the numeric perspective. Based on these results, we are confident that it can stand the test in time and replication, and the proposed approach would form a fundamental module to deal with the temporal issues in SNN training given its simplicity and effectiveness.

Thus, we believe that our work is a very solid and inspiring piece for the SNN community especially those working on dynamic vision data. We wholeheartedly hope that the reviewers can re-consider its potential contribution to the SNN field and re-evaluate our work.

Sincerely yours,

ICLR 2022 Conference Paper1430 Authors

---

### Decision · Program_Chairs · 2022-01-20

**Decision:**

Accept (Poster)

**Comment:**

The paper proposes a new loss function for the training of spiking neural networks leading to significant improvements in generalization performance across a variety of datasets and network architectures. While conceptually simple, the approach leads to substantial performance gains, and some intuition is provided to explain its success.

The reviewers are split on the issue of significance of the paper, in part due to the simplicity of the proposed loss function. Still, good results speak for themselves, and the effectiveness of the technique has been demonstrated thoroughly.